# Gradient Estimation for Unseen Domain Risk Minimization with Pre-Trained Models

## Abstract

Domain generalization aims to build generalized models that perform well on unseen domains when only source domains are available for model optimization. Recent studies have demonstrated that large-scale pre-trained models could play an important role in domain generalization by providing their *generalization power*. However, large-scale pre-trained models are not fully equipped with target task-specific knowledge due to a discrepancy between the pre-training objective and the target task. Although the task-specific knowledge could be learned from source domains by *fine-tuning*, this hurts the generalization power of the pre-trained models because of *gradient bias* toward the source domains. To address this issue, we propose a new domain generalization method that estimates *unobservable gradients* that reduce potential risks in unseen domains, using a large-scale pre-trained model. Our proposed method allows the pre-trained model to learn task-specific knowledge further while preserving its generalization ability with the estimated gradients. Experimental results show that our proposed method outperforms baseline methods on DOMAINBED, a standard benchmark in domain generalization. We also provide extensive analyses to demonstrate that the estimated unobserved gradients relieve the gradient bias, and the pre-trained model learns the task-specific knowledge without sacrificing its generalization power.

## 1 Introduction

Many machine learning studies assume that training and test data are independent and identically distributed (*i.i.d*). However, this *i.i.d* assumption does not always hold in real-world scenarios where distribution shifts between training and test data occur frequently. Thus, traditional machine learning models often show poor performance on unseen domains shifted from source (training) domains (Quinonero-Candela et al., 2008; Torralba & Efros, 2011). To tackle this problem, *domain generalization* has attracted much attention recently.

The main goal of domain generalization is to build generalized models that also perform the target task (*e.g.,* classification) well on unseen domains (*e.g.,* painted images) when only source domains (*e.g.,* realistic images) are accessible during model optimization. Early domain generalization studies (Muandet et al., 2013; Ganin et al., 2016; Li et al., 2018b) have focused on learning domain-invariant representations across the source domains. However, Gulrajani & Lopez-Paz (2021) have recently shown that simple empirical risk minimization (ERM) (Vapnik, 1999) outperforms the previous methods on DOMAINBED, a benchmark for domain generalization, with pre-trained ResNet-50 (He et al., 2016). Moreover, Yu et al. (2021) provide empirical evidence that large-scale pre-trained models could play an important role in domain generalization by providing their generalization power.

Motivated by this, several studies have begun to leverage the generalization power of large-scale pre-trained models. Cha et al. (2022) employ a pre-trained model for regularization, considering it as an approximation of the oracle model on any domain, and Li et al. (2022) utilize a frozen pre-trained model as a feature extractor. These studies have proven the usefulness of pre-trained models in domain generalization. However, the pre-trained models used in those studies cannot learn task-specific knowledge further since they are frozen during model optimization to preserve their generalization ability. To learn the task-specific knowledge, one can choose *fine-tuning* that updates all the parameters of pre-trained models by optimizing the models on the source domains.

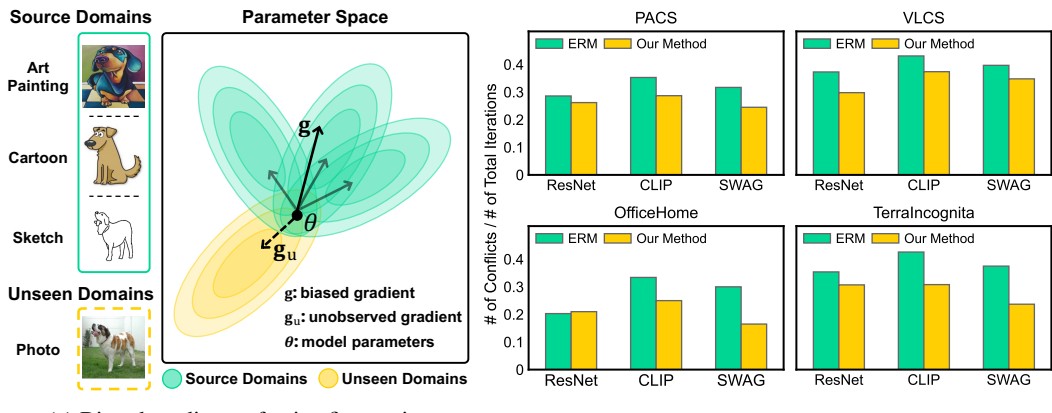

(a) Biased gradients of naive fine-tuning    (b) Gradient conflict between $\mathbf{g}$ and $\mathbf{g}_u$

Figure 1: **(a):** model optimization is influenced by the gradient $\mathbf{g}$ biased toward the source domains, neglecting the unobservable gradient $\mathbf{g}_u$ that could minimize risks in the unseen domains. **(b):** Gradient "*conflicts*" (Yu et al., 2020; Mansilla et al., 2021) between $\mathbf{g}$ and $\mathbf{g}_u$ (i.e., $\mathbf{g} \cdot \mathbf{g}_u < 0$) constantly occur throughout the whole fine-tuning iterations due to the gradient bias. Our proposed method reduces the number of gradient conflicts by adding the estimated unobservable gradient $\tilde{\mathbf{g}}_u$ to the biased gradient $\mathbf{g}$. This observation indicates that the gradient bias is relieved with the estimated gradient during model optimization. The more details are described in § 3.4.

However, Kumar et al. (2022) demonstrate that fine-tuning distorts generalized representations of the pre-trained models. Namely, fine-tuning hurts the generalization ability of pre-trained models.

In this paper, we interpret the above issue in terms of *gradient bias* during model optimization. As shown in Figure 1a, the gradient of naive fine-tuning is biased toward the source domains because it is computed by only the source domains, disregarding unseen domains. Although this biased gradient reduces empirical risks in the source domains with the learning of task-specific knowledge, it probably increases risks in the unseen domains. We argue that the gradient bias would be relieved if gradients that lower the risks in the unseen domains are observable.

To this end, we propose a new domain generalization method, called *GESTUR*, which estimates the unobservable gradients with a large-scale pre-trained model. GESTUR consists of two key components: a task expert (TE) and a generalization expert (GE). Based on ERM where gradients tend to be biased to the source domains, TE learns task-specific knowledge from source domains directly to transfer the knowledge to GE. Meanwhile, GE learns the task-specific knowledge from TE indirectly via exponential moving average (EMA) while preserving the generalization ability of a large-scale pre-trained model. Still, the gradient bias of TE might impair the generalization ability of GE. To mitigate this, GE is utilized to estimate the unobservable gradient that minimizes risks in unseen domains for TE based on the assumption that large-scale pre-trained models could act as a loose approximation of the oracle model of unseen domains (§ 2). As shown in Figure 1b, the biased gradient of TE is relieved by simply adding the estimated unobservable gradient to the biased gradient, improving domain generalization performance (§ 3). Extensive experiments and analyses demonstrate that GESTUR outperforms baseline methods by learning the task-specific knowledge appropriately from source domains while preserving the generalization ability of large-scale pre-trained models.

**Contributions:** (1) We propose a simple yet effective domain generalization method that learns task-specific knowledge while preserving the generalization ability of large-scale pre-trained models. Our proposed method estimates the unobservable gradients that reduce potential risks in unseen domains to relieve the gradient bias toward source domains, based on the two experts, TE and GE. (2) We conduct extensive experiments to show the effectiveness of our proposed method in domain generalization. By providing careful analyses, we demonstrate that the unobservable gradients could be estimated with a large-scale pre-trained model, and it relieves the gradient bias. We also demonstrate that our proposed method learns task-specific knowledge without sacrificing the generalization ability of the large-scale pre-trained model.

## 2 METHODOLOGY

### 2.1 PRELIMINARIES

**Problem formulation.** Let $\mathcal{D}_s$ and $\mathcal{D}_u$ be sets of source domains and unseen domains, respectively. Each domain $\mathcal{D}$ contains the total number of $n_\mathcal{D}$ data samples, $\{(x_i, y_i)\}_{i=1}^{n_\mathcal{D}} \sim \mathcal{D}$, where each data sample $(x_i, y_i)$ consists of an input $x_i$ and its target label $y_i$. The $n_\mathcal{D}$ data samples are *i.i.d* over some probability distribution. The main goal of domain generalization is to build a model $\theta$ that performs well on the unseen domains $\mathcal{D}_u$ when the source domains $\mathcal{D}_s$ are only available:

$$\min_\theta \mathbb{E}_{\mathcal{D} \sim \mathcal{D}_u} \mathbb{E}_{(x,y) \sim \mathcal{D}}[\ell((x, y); \theta)], \tag{1}$$

where $\ell((x, y); \theta)$ is the loss function defined for the model $\theta$ on the data sample $(x, y)$. Note that this study focuses on solving classification tasks. Hence, we denote the model in detail as $\theta = \{\theta^f; \theta^c\}$ consisting of its feature extractor $\theta^f$ and classifier $\theta^c$.

**Motivation.** With success in many downstream tasks, it has become a convention to initialize the feature extractor $\theta^f$ with a large-scale pre-trained model. Although pre-trained models provide better feature representations than randomly initialized parameters, they do not fully equip task-specific knowledge yet. It is because there is a discrepancy between the pre-training objective and the target task. For example, CLIP (Radford et al., 2021) is pre-trained to match web-crawled image-caption pairs, whereas the target task is to classify data into seven classes (*e.g.,* horse and dog), in the case of PACS (Li et al., 2017). Therefore, many studies have adopted fine-tuning that updates all the parameters of the feature extractor $\theta^f$ to learn the task-specific knowledge by optimizing the model on source domains $\mathcal{D}_s$. However, Kumar et al. (2022) observe that fine-tuning impairs generalization ability of pre-trained models during the learning of task-specific knowledge.

We try to interpret this issue at the gradient level. Based on ERM (Vapnik, 1999), the gradient **g** of fine-tuning is computed for the model $\theta$ on the source domains $\mathcal{D}_s$, as follows:

$$\mathbf{g} = \nabla_\theta \mathbb{E}_{(x,y) \sim \mathcal{B}}[\ell((x, y); \theta)], \tag{2}$$

where $\mathcal{B}$ is a mini-batch sampled from the source domains $\mathcal{D}_s$. The gradient **g** is influenced by only the source domains $\mathcal{D}_s$ because the unseen domains $\mathcal{D}_u$ are not accessible. Namely, the gradient is biased toward the source domains. We presume that this gradient bias degrades generalization performance in the unseen domains.

### 2.2 GESTUR: GRDIENT ESTIMATION FOR UNSEEN DOMAIN RISK MINIMIZATION WITH PRE-TRAINED MODELS

We hypothesize that the gradient bias mentioned above could be relieved if the unobservable gradient $\mathbf{g}_u$ minimizing risks in the unseen domains is computable. To achieve this, we borrow the assumption of Cha et al. (2022) that large-scale pre-trained models are the approximation of the oracle model $\theta^*$ which is optimally generalized for any domain $\mathcal{D}$. Since the unobservable gradient $\mathbf{g}_u$ cannot be computed from the unseen domains $\mathcal{D}_u$ directly, we consider the direction from the current model $\theta$ to the oracle model $\theta^*$ as the unobservable gradient $\mathbf{g}_u$. However, the oracle model is inaccessible in practice. Hence, we estimate the unobservable gradient using a large-scale pre-trained model as the approximation of the oracle model.

Note that we aim to estimate the unobservable gradient $\mathbf{g}_u$ for the unseen domains $\mathcal{D}_u$ to alleviate the gradient bias, so the above assumption needs to be more elaborate due to the following reasons. First, we intend to design the unobservable gradient for the unseen domains only rather than any domain. Second, pre-trained models do not have task-specific knowledge yet, as described in § 2.1. Therefore, we slightly modify the assumption as follows: pre-trained models are the *loose* approximation of the oracle model $\theta_u^*$ of *the unseen domains* $\mathcal{D}_u$, and they could get *closer* to the oracle model by learning task-specific knowledge. Based on this assumption, we propose a simple yet effective domain generalization method, GESTUR, which estimates the unobservable gradient $\mathbf{g}_u$ for unseen domain risk minimization with a large-scale pre-trained model.

**Task expert and generalization expert.** GESTUR consists of two classification models: a task expert (TE, $\theta_{\text{TE}}$) and a generalization expert (GE, $\theta_{\text{GE}}$), which are complementary to each other.

Their feature extractors are both initialized with a large-scale pre-trained model $\theta_0$, respectively. TE learns task-specific knowledge from the source domains $\mathcal{D}_s$ directly to transfer the knowledge to GE. Meanwhile, GE also learns task-specific knowledge from TE via EMA, but it preserves the generalization ability of the pre-trained model deliberately. Here, the gradient bias of TE might hurt the generalization ability of GE because the knowledge of TE is injected into GE. To relieve the gradient bias, GE is used to estimate the unobservable gradient $\mathbf{g}_u$ as the loose approximation of the oracle model $\theta_u^*$ for the unseen domains $\mathcal{D}_u$. Our proposed GESTUR is summarized in Algorithm 1.

**Gradient estimation.** Using Equation 2, the gradient $\mathbf{g}$ for TE is computed as $\mathbf{g} = \nabla_\theta \mathbb{E}_{(x,y)\sim\mathcal{B}}[\ell((x,y); \theta_{\text{TE}})]$ while learning task-specific knowledge. The gradient $\mathbf{g}$ is biased toward the source domains $\mathcal{D}_s$. A gradient that minimizes risks in the unseen domains could relieve the gradient bias, but it is unobservable. When we have access to the oracle model $\theta_u^*$ of the unseen domains $\mathcal{D}_u$, we can direct the current model to head to the oracle model instead of empirically calculating the unobservable gradient from the unseen domains. Hence, we treat the direction from the current model $\theta_{\text{TE}}$ to the oracle model $\theta_u^*$ as the unobservable gradient $\mathbf{g}_u$:

---

**Algorithm 1** GESTUR

1: **Input:** task expert $\theta_{\text{TE}}$, generalization expert $\theta_{\text{GE}}$, gradient scale factor $\lambda$, and moving average coefficient $m$.
2: **Init:** initialize the feature extractors $\theta_{\text{TE}}^f$ and $\theta_{\text{GE}}^f$ with a pre-trained model $\theta_0$ and randomly initialize the classifiers $\theta_{\text{TE}}^c$ and $\theta_{\text{GE}}^c$.
3: **Output:** the updated generalization expert $\theta_{\text{GE}}$
4: **for** sampled mini-batch $\mathcal{B}$ from the source domains $\mathcal{D}_s$ **do**
5:      $\mathbf{g} = \nabla_\theta \mathbb{E}_{(x,y)\sim\mathcal{B}}[\ell((x,y); \theta_{\text{TE}})]$
6:      $\tilde{\mathbf{g}}_u^f = \theta_{\text{GE}}^f - \theta_{\text{TE}}^f$
7:      $\tilde{\mathbf{g}}_u^f = \lambda \|\mathbf{g}^f\|_2 \cdot \dfrac{\tilde{\mathbf{g}}_u^f}{\|\tilde{\mathbf{g}}_u^f\|_2}$
8:      $\mathbf{g}^f = (\mathbf{g}^f + \tilde{\mathbf{g}}_u^f)/2$
9:      update $\theta_{\text{TE}}^f$ with $\mathbf{g}^f$ and update $\theta_{\text{TE}}^c$ with $\mathbf{g}^c$
10:     update $\theta_{\text{GE}} = m\theta_{\text{GE}} + (1 - m)\theta_{\text{TE}}$
11: **end for**

---

$$\mathbf{g}_u = \theta_u^* - \theta_{\text{TE}}. \tag{3}$$

In fact, it is infeasible to access the oracle model. Thus, we estimate the unobservable gradient using GE that approximates the oracle model loosely, as follows:

$$\tilde{\mathbf{g}}_u = \theta_{\text{GE}} - \theta_{\text{TE}}. \tag{4}$$

This estimated gradient $\tilde{\mathbf{g}}_u$ is used to relieve the gradient bias during the parameter optimization.

**Parameter optimization.** We want to emphasize again that GESTUR leverages the generalization power of large-scale pre-trained models to relieve the gradient bias which distorts the generalized feature representations of the feature extractor $\theta^f$. Hence, we limit the scope of usage of the estimated unobservable gradient $\tilde{\mathbf{g}}_u$ only to the feature extractor $\theta^f$, not the classifier $\theta^c$.

For TE, the estimated gradient $\tilde{\mathbf{g}}_u^f$ for the feature extractor $\theta_{\text{TE}}^f$ is added to the biased gradient $\mathbf{g}^f$ for the same feature extractor, as follows:

$$\mathbf{g}^f = \frac{1}{2}\Big(\mathbf{g}^f + \lambda \|\mathbf{g}^f\|_2 \cdot \frac{\tilde{\mathbf{g}}_u^f}{\|\tilde{\mathbf{g}}_u^f\|_2}\Big), \tag{5}$$

where $\lambda$ is a gradient scale factor that controls the influence of the normalized $\tilde{\mathbf{g}}_u^f$. The feature extractor $\theta_{\text{TE}}^f$ is updated with the gradient $\mathbf{g}^f$ adjusted by $\tilde{\mathbf{g}}_u^f$. On the other hand, the classifier $\theta_{\text{TE}}^c$ of TE is updated with its original gradient $\mathbf{g}^c$.

As our assumption, GE can get closer to the oracle model $\theta^*$ by learning task-specific knowledge. However, the generalization ability of GE decreases when we optimize GE on the source domains $\mathcal{D}_s$ directly to learn the task-specific knowledge. Therefore, we inject the learned task-specific knowledge of TE into GE delicately via EMA:

$$\theta_{\text{GE}} = m\theta_{\text{GE}} + (1 - m)\theta_{\text{TE}}, \tag{6}$$

where $m$ is the moving average coefficient. By encouraging the parameters of GE to change slowly, EMA is helpful in preserving the generalization ability of GE. Since the goal of domain generalization is to build a model that minimizes the risk of the unseen domains, we choose GE, designed to approximate the oracle model of the domains, as our final model $\theta$.

Table 1: Evaluation results (%) on the five datasets with the three different pre-trained models.

| Method | PACS | VLCS | OfficeHome | TerraInc | DomainNet | Avg. |
|--------|------|------|-----------|----------|-----------|------|
| *Using ResNet-50 pre-trained on ImageNet.* | | | | | | |
| ERM | 84.2 ±0.1 | 77.3 ±0.1 | 67.6 ±0.2 | 47.8 ±0.6 | 44.0 ±0.1 | 64.2 |
| SagNet | 86.3 ±0.2 | 77.8 ±0.5 | 68.1 ±0.1 | 48.6 ±1.0 | 40.3 ±0.1 | 64.2 |
| SelfReg | 85.6 ±0.4 | 77.8 ±0.9 | 67.9 ±0.7 | 47.0 ±0.3 | 42.8 ±0.0 | 64.2 |
| CORAL | 86.2 ±0.3 | 78.8 ±0.6 | 68.7 ±0.3 | 47.6 ±1.0 | 41.5 ±0.1 | 64.5 |
| mDSDI | 86.2 ±0.2 | 79.0 ±0.3 | 69.2 ±0.4 | 48.1 ±1.4 | 42.8 ±0.1 | 65.1 |
| GVRT | 85.1 ±0.3 | 79.0 ±0.2 | 70.1 ±0.1 | 48.0 ±1.4 | 44.1 ±0.1 | 65.2 |
| MIRO | 85.4 ±0.4 | 79.0 ±0.0 | 70.5 ±0.4 | 50.4 ±1.1 | 44.3 ±0.2 | 65.9 |
| SMA | 87.5 ±0.2 | 78.2 ±0.2 | 70.6 ±0.1 | 50.3 ±0.5 | 46.0 ±0.1 | 66.5 |
| SWAD | **88.1** ±0.1 | 79.1 ±0.1 | 70.6 ±0.2 | 50.0 ±0.3 | **46.5** ±0.1 | 66.9 |
| **GESTUR** | 88.0 ±0.2 | **80.1** ±0.2 | **71.1** ±0.0 | **51.3** ±0.2 | 46.3 ±0.1 | **67.4** |
| *Using ViT-B/16 with CLIP.* | | | | | | |
| ERM | 83.4 ±0.5 | 75.9 ±1.3 | 66.4 ±0.5 | 35.3 ±0.8 | 44.4 ±0.6 | 61.1 |
| SWAD | 91.3 ±0.1 | 79.4 ±0.4 | 76.9 ±0.1 | 45.4 ±0.5 | 51.7 ±0.8 | 68.9 |
| MIRO | 95.6 ±0.8 | 82.2 ±0.3 | 82.5 ±0.1 | 54.3 ±0.4 | 54.0 ±0.3 | 73.7 |
| **GESTUR** | **96.0** ±0.0 | **82.8** ±0.1 | **84.2** ±0.1 | **55.7** ±0.2 | **58.9** ±0.1 | **75.5** |
| *Using RegNetY-16GF with SWAG.* | | | | | | |
| ERM | 89.6 ±0.4 | 78.6 ±0.3 | 71.9 ±0.6 | 51.4 ±1.8 | 48.5 ±0.6 | 68.0 |
| SWAD | 94.7 ±0.2 | 79.7 ±0.2 | 80.0 ±0.1 | 57.9 ±0.7 | 53.6 ±0.6 | 73.2 |
| MIRO | **97.4** ±0.2 | 79.9 ±0.6 | 80.4 ±0.2 | 58.9 ±1.3 | 53.8 ±0.1 | 74.1 |
| SMA | 95.5 ±0.0 | 80.7 ±0.1 | 82.0 ±0.0 | 59.7 ±0.0 | 60.0 ±0.0 | 75.6 |
| **GESTUR** | 96.9 ±0.1 | **83.5** ±0.1 | **83.1** ±0.0 | **61.1** ±0.4 | **60.1** ±0.0 | **76.9** |

## 3 EXPERIMENTS

### 3.1 EXPERIMENTAL SETUP

**Datasets.** We conduct experiments using five popular domain generalization benchmark datasets: PACS (Li et al., 2017) (4 domains & 7 classes), VLCS (Fang et al., 2013) (4 domains & 5 classes), OfficeHome (Venkateswara et al., 2017) (4 domains & 65 classes), TerraIncognita (Beery et al., 2018) (4 domains & 10 classes), and DomainNet (Peng et al., 2019) (6 domains & 345 classes).

**Pre-trained models.** GESTUR heavily relies on pre-trained models. Therefore, we employ three pre-trained models of different sizes to verify that the proposed method performs well with various pre-trained models generally: ResNet-50 (He et al., 2016) pre-trained on ImageNet (Deng et al., 2009) (RN50), ViT-B/16 (Dosovitskiy et al., 2021) with CLIP (Radford et al., 2021) (CLIP), and RegNetY-16GF (Radosavovic et al., 2020) with SWAG (Singh et al., 2022) (SWAG).

**Evaluation protocol.** We adopt the experimental protocol of DOMAINBED, which enforces fair and realistic evaluations (*e.g.,* same model selection criterion) across competitors. We divide the data from each domain into 80% and 20% splits and follow *training-domain validation set* strategy for the model selection and the hyperparameter search in every experiment. We also repeat every experiment three times to reduce the randomness in dataset splits and parameter initialization, similar to Gulrajani & Lopez-Paz (2021), and report the mean and standard error of the experimental results.

**Implementation details.** Our implementation is built on the codebase of Cha et al. (2022). We use Adam optimizer (Kingma & Ba, 2015) for parameter optimization. GESTUR has two hyperparameters, the gradient scale factor ($\lambda$) and the moving average coefficient ($m$). In every experiment, we search the optimal $\lambda$ from $\{0.01, 0.05, 0.1, 0.5\}$ and fix $m$ as 0.999. Other hyperparameters such as learning rate, weight decay, and dropout are searched in the same way as Cha et al. (2022). We explain more details of implementation in Appendix A.

Table 2: Evaluation results (%) on the four datasets with the three different pre-trained models. We separate the cases where GESTUR uses TE and GE as the final model, respectively.

| Method | PACS | VLCS | OfficeHome | TerraInc | Avg. |
|---|---|---|---|---|---|
| *Using ResNet-50 pre-trained on ImageNet.* | | | | | |
| ERM | 84.2 ±0.1 | 77.3 ±0.1 | 67.6 ±0.2 | 47.8 ±0.6 | 69.2 |
| GESTUR w/ TE | 84.9 ±0.1 | 79.2 ±0.5 | 66.3 ±0.2 | 45.6 ±1.3 | 69.0 |
| **GESTUR w/ GE** | **88.0** ±0.2 | **80.1** ±0.2 | **71.1** ±0.0 | **51.3** ±0.2 | **72.6** |
| *Using ViT-B/16 with CLIP.* | | | | | |
| ERM | 83.4 ±0.5 | 75.9 ±1.3 | 66.4 ±0.5 | 35.3 ±0.8 | 65.3 |
| GESTUR w/ TE | 90.7 ±0.9 | 82.4 ±0.4 | 76.9 ±0.5 | 50.4 ±0.2 | 75.1 |
| **GESTUR w/ GE** | **96.0** ±0.1 | **82.8** ±0.1 | **84.2** ±0.1 | **55.7** ±0.2 | **79.7** |
| *Using RegNetY-16GF with SWAG.* | | | | | |
| ERM | 89.6 ±0.4 | 78.6 ±0.3 | 71.9 ±0.6 | 51.4 ±1.8 | 72.9 |
| GESTUR w/ TE | 94.8 ±0.5 | 82.5 ±0.4 | 77.7 ±0.2 | 54.7 ±2.0 | 77.4 |
| **GESTUR w/ GE** | **96.9** ±0.1 | **83.5** ±0.1 | **83.1** ±0.0 | **61.1** ±0.4 | **81.2** |

**Baselines.** We exhaustively compare our proposed method with various baseline methods in the experiment. For simplicity, we report only the experimental results of baseline methods that show the higher performance than ERM (Vapnik, 1999), the simplest baseline method. We describe the baseline methods and report the full version of the results in Appendix B.1.

## 3.2 MAIN RESULTS

**Results on `RN50`.** The first part of Table 1 shows the experimental results where `RN50` is used to initialize the feature extractor. GESTUR achieves the best performance for all the datasets except DomainNet. In detail, the proposed method outperforms ERM by an average of 3.2%$p$. Furthermore, our proposed method improves the runner-up by: 1.0%$p$ in VLCS, 0.5%$p$ in OfficeHome, and 1.3%$p$ in TerraIncognita. Especially, the proposed method outperforms the state-of-the-art method (SWAD (Cha et al., 2021)) by an average of 0.5%$p$.

**Results on `CLIP` and `SWAG`.** In the second and third parts of Table 1, we show the experimental results where the larger pre-trained models, `CLIP` and `SWAG`, are used to initialize the feature extractor, respectively. In summary, GESTUR achieves the best performance in all the datasets. In detail, the proposed method outperforms MIRO that also leverages generalization power of large-scale pre-trained models by 1.8%$p$ and 2.8%$p$ on `CLIP` and `SWAG`, respectively. From this, we verify that the proposed method successfully leverages the generalization ability of pre-trained models compared to other baseline methods. Interestingly, we observe that the performance gap between the proposed method and ERM increases as the size of the pre-trained model increases.

## 3.3 COMPARISON BETWEEN THE TASK EXPERT AND THE GENERALIZATION EXPERT

**Setup.** GESTUR consists of two essential components: the task expert (TE) and the generalization expert (GE). In this paper, we use GE as the final model based on the assumption that GE is set as the approximation of the oracle model of unseen domains. Nevertheless, TE is also designed to preserve the generalization ability of pre-trained models since it also considers the estimated unobservable gradient in every update to relieve its gradient bias. Therefore, we compare the performance of ERM, GESTUR w/ GE, and its variant GESTUR w/ TE based on the hyperparameters searched in § 3.2 to show that they preserve the generalization ability of large-scale pre-trained models.

**Results.** As shown in Table 2, GESTUR w/ GE achieves the best performance in all experiments. Also, GESTUR w/ TE outperforms ERM by averages of 9.8%$p$ and 4.5%$p$ when using `CLIP` and `SWAG`, respectively. The performance of GESTUR w/ TE is higher when the larger pre-trained models are given, similar to the observation in § 3.2. These observations demonstrate that GESTUR w/

TE could preserve the generalization ability of the pre-trained models with the estimated unobservable gradient, *i.e.,* the gradient bias of TE is relieved. Moreover, GESTUR w/ GE shows a higher performance than GESTUR w/ TE, which indicates that EMA ensures the model preserves generalization ability during the learning of task-specific knowledge stable. From these, we reaffirm the justification for our choice of GE as the final model.

### 3.4 COMPARISON WITH ERM IN TERMS OF GRADIENT BIAS

**Setup.** As described in § 1, we suspect that the gradient bias degrades the domain generalization performance. We further conduct analysis to check how much gradient bias occurs during the fine-tuning and how much gradient bias is alleviated by our proposed method. To quantify the gradient bias, we borrow the concept of *gradient conflict* (Yu et al., 2020; Mansilla et al., 2021): there is a conflict between two gradients $\mathbf{g}_i$ and $\mathbf{g}_j$ if $\mathbf{g}_i \cdot \mathbf{g}_j < 0$. For every iteration, we first sample two mini-batches from both source domains and an unseen domain, respectively. We then compute losses of the mini-batches, and calculate gradients $\mathbf{g}$ and $\mathbf{g}_u$ from the losses, respectively. Finally, we count the

Table 3: The percentage (%) of gradient conflicts between $\mathbf{g}$ and $\mathbf{g}_u$ to the whole training iterations.

| Method | PACS | VLCS | OH | TI | Avg. |
|---|---|---|---|---|---|
| *Using ResNet-50 pre-trained on ImageNet.* | | | | | |
| ERM | 28.6 | 37.3 | **20.3** | 35.4 | 30.4 |
| **GESTUR** | **26.2** | **29.8** | 21.0 | **30.7** | **26.9** |
| *Using ViT-B/16 with CLIP.* | | | | | |
| ERM | 35.3 | 43.1 | 33.4 | 42.6 | 38.6 |
| **GESTUR** | **28.7** | **37.4** | **25.0** | **30.8** | **30.5** |
| *Using RegNetY-16GF with SWAG.* | | | | | |
| ERM | 31.7 | 39.7 | 30.0 | 37.5 | 34.7 |
| **GESTUR** | **24.5** | **34.8** | **16.5** | **23.7** | **24.9** |

number of iterations where the gradient conflict ($\mathbf{g} \cdot \mathbf{g}_u < 0$) occurs, for ERM and GESTUR. Here, we update the model using only the gradient $\mathbf{g}$ since unseen domains are inaccessible in practice.

**Results.** As shown in Table 3, GESTUR reduces the gradient conflicts of ERM by around 11.5%, 21%, and 28.2% for the pre-trained models, respectively. From this, we verify that our proposed method relieves gradient bias by estimating unobservable gradients with the pre-trained model. We observe that gradient conflicts occur more often in GESTUR than ERM on only the experimental setup (OfficeHome w/ `RN50`), which is consistent with the performance in Table 2 where ERM outperforms GESTUR w/ TE. This observation indicates that the domain generalization performance is affected by the gradient bias represented as the gradient conflicts in this analysis. Additional analysis on the similarity of the true and estimated unobservable gradients is provided in Appendix C.3.

### 3.5 TASK-SPECIFIC KNOWLEDGE LEARNED BY THE GENERALIZATION EXPERT

Table 4: Linear probing performance (%) with the two different pre-trained feature extractors: *frozen* pre-trained model $\theta_0$ and the feature extractor $\theta_{\mathrm{GE}}^f$ of GE.

| Model | PACS | VLCS | OfficeHome | TerraInc | Avg. |
|---|---|---|---|---|---|
| *Using ViT-B/16 with CLIP.* | | | | | |
| *frozen* | 98.5 ±0.1 | 88.5 ±0.2 | 89.3 ±0.1 | 83.4 ±0.2 | 89.9 |
| GE | **98.7** ±0.1 | **90.0** ±0.6 | **89.4** ±0.3 | **88.3** ±0.1 | **91.6** |
| *Using RegNetY-16GF with SWAG.* | | | | | |
| *frozen* | **98.9** ±0.1 | 87.1 ±0.2 | 89.6 ±0.0 | 89.3 ±0.0 | 91.2 |
| GE | 98.7 ±0.1 | **88.8** ±0.2 | **90.1** ±0.2 | **90.2** ±0.1 | **92.0** |

**Setup.** We conduct additional experiments to show that the feature extractor $\theta_{\mathrm{GE}}^f$ of GE learns task-specific knowledge successfully. Linear probing that updates parameters of only the classifier while freezing those of the feature extractor is common practice for assessing representation quality. We assume that the more task-specific knowledge the feature extractor learns, the better linear probing performance it exhibits in unseen domains targeting the same task. In detail, we first train GE on source domains and then evaluate linear probing performance on an unseen domain with the trained feature extractor of GE. We compare it with the case that a *frozen* pre-trained model is used as the

feature extractor. For linear probing, we simply train a logistic regression classifier on the output feature representations of each feature extractor using the unseen domain only. Note that, in this analysis, we use `CLIP` and `SWAG` which are pre-trained with objectives significantly different from the target task to demonstrate the effectiveness of the newly learned task-specific knowledge clearly.

**Results.** As shown in Table 4, GE outperforms *frozen* in all benchmark datasets except the one case where the two models reach the near 99% performance. This shows that GE is learning task-specific knowledge further during training, which makes it a better approximation of the oracle model. The result supports our claim that pre-trained models are not fully equipped with target task-specific knowledge, and injecting the knowledge further increases performance.

## 3.6 RELATIONSHIP BETWEEN $\lambda$ AND THE SIZE OF THE PRE-TRAINED MODEL

Table 5: Evaluation results (%) on PACS with the three different pre-trained models varying $\lambda$.

| Dataset (size) | Pre-training | Architecture | $\lambda$ | | | |
|---|---|---|---|---|---|---|
| | | | 0.01 | 0.05 | 0.1 | 0.5 |
| ImageNet (1.3M) | ERM | ResNet-50 | **88.0** ±0.2 | 86.0 ±0.2 | 82.1 ±0.2 | 73.4 ±0.4 |
| CLIP (400M) | CLIP | ViT-B/16 | 94.8 ±0.2 | 96.0 ±0.0 | **96.2** ±0.1 | 96.0 ±0.0 |
| Instagram (3.6B) | SWAG | RegNetY-16GF | 96.3 ±0.2 | 96.9 ±0.1 | 97.6 ±0.1 | **97.9** ±0.1 |

**Setup.** Our proposed GESTUR controls the scale of the estimated unobservable gradients that reduce risks in unseen domains using the gradient scale factor $\lambda$. To verify the effect of the scale factor, we observe the performance change varying the scale factor.

**Results.** In Table 5, `RN50` achieves the best performance with $\lambda = 0.01$. On the other hand, the larger pre-trained models, `CLIP` and `SWAG` achieve the best performance with the relatively larger $\lambda = 0.1$ and $\lambda = 0.5$, respectively. We summarize more results on other datasets (*i.e.,* VLCS, OfficeHome, and TerraIncognita) in Appendix C.1, and they show the similar pattern as in PACS.

Intuitively, the larger pre-trained models act as a better approximation of the oracle model than the small one because they are likely to encounter various domains from the huge web-crawled datasets during pre-training. They help to estimate unobservable gradients more accurately. The larger gradient scale factor, gradients $\mathbf{g}$ of TE is more affected by the estimated unobservable gradients $\tilde{\mathbf{g}}_u$ while optimizing the model on source domains. From this, we can conclude that the larger scale factor improves the generalization performance when larger pre-trained models are given.

# 4 RELATED WORK

## 4.1 DOMAIN GENERALIZATION

**Domain alignment.** Domain alignment is to learn domain-invariant feature representations by removing domain-specific knowledge in the representations. Adversarial training is widely adopted to learn domain invariant features through a min-max game between a feature extractor and a domain discriminator (Ganin et al., 2016; Li et al., 2018c; Matsuura & Harada, 2020; Zhu et al., 2022). On the other hand, several studies (Muandet et al., 2013; Sun & Saenko, 2016; Li et al., 2018b) aim to minimize feature divergence across source domains. Recently, contrastive learning-based algorithms (Kim et al., 2021; Yao et al., 2022) have been proposed to minimize distances between feature representations of samples in the same class, regardless their domains.

**Data augmentation.** Many studies have employed data augmentation techniques to improve domain generalization performance. For example, Gulrajani & Lopez-Paz (2021) apply simple data augmentation techniques as a default setup in DOMAINBED and some studies (Wang et al., 2020; Xu et al., 2020; Yan et al., 2020) utilize Mixup (Zhang et al., 2017). Recently, a few works (Zhou et al., 2021; Nam et al., 2021; Kang et al., 2022) focus on image style, based on the idea that domain gap is closely related to image style. On the other side, some works on single domain generalization

introduce adversarial data augmentation (Volpi et al., 2018; Fan et al., 2021; Qiao et al., 2020) to generate hard samples adversarially while assuring their reliability.

**Gradient-based.** Recently, several studies utilize gradients to build generalized models, especially by aligning gradients from different domains. Mansilla et al. (2021) exploit gradient agreement for gradient surgery, based on the hypothesis that conflicting gradients contain domain-specific information. Shi et al. (2022) propose a training method that maximizes inner product between source domain gradients to match optimization paths across domains. Similarly, Rame et al. (2022) try to match domain-level Hessian to align loss landscapes across domains. As another line of work, Huang et al. (2020) introduce the self-challenging algorithm that iteratively masks dominant features, which are selected by the scale of the gradients.

**Meta-learning-based.** Since simulating domain shift by dividing source domains into meta-train and meta-test domains was first introduced in MLDG (Li et al., 2018a), several approaches have been proposed in a similar setting. For example, Balaji et al. (2018) propose to learn a regularizer for classifier weights and Zhang et al. (2021a) bring the idea of Reptile (Nichol et al., 2018) to MLDG to further increase performance with a multi-view framework. On the other hand, Zhang et al. (2021b) employ meta-learning to adaptively predict model parameters from a batch of inputs.

**Others.** Some of the works bring concepts of causality (Lv et al., 2022), optimize the worst-case performance (Sagawa et al., 2019; Krueger et al., 2021), utilize text labels (Min et al., 2022), or average model weights from different epochs (Cha et al., 2021; Arpit et al., 2022).

Our work differs from aforementioned approaches in that we mainly concentrate on effectively utilizing large-scale pre-trained models.

## 4.2 DOMAIN GENERALIZATION WITH PRE-TRAINED MODELS

Recently, Gulrajani & Lopez-Paz (2021) empirically show that simple ERM (Vapnik, 1999) outperforms most of early methods with pre-trained ResNet-50 (He et al., 2016). Yu et al. (2021) show that using large-scale models pre-trained on massive datasets improves out-of-distribution performance. Kumar et al. (2022) find that fine-tuning distorts pre-trained features and propose the *linear-probing then fine-tuning* to mitigate the feature distortion. Wortsman et al. (2022) find that linearly interpolating the zero-shot and fine-tuned parameters of a pre-trained model improves performance in both source and unseen domains. Although GESTUR's EMA (Equation 6) looks similar to their interpolation, GESTUR updates the pre-trained model to inject task-specific knowledge. Li et al. (2022) propose a method to efficiently leverage a pool of large-scale pre-trained models through specialty-aware ensemble learning. Cha et al. (2022) propose MIRO, a regularization method that targets to minimize mutual information with pre-trained models which approximate the oracle model. In this work, we share similar motivation with MIRO in that we initially approximate the oracle model with a large-scale pre-trained model. However, we iteratively inject task-specific knowledge into the approximation of the oracle model, resulting in a better approximation.

## 5 CONCLUSION AND FUTURE WORK

In this paper, we propose a new domain generalization method that learns task-specific knowledge while preserving the generalization ability of large-scale pre-trained models. We point out that gradient bias toward source domains hurts the generalization ability of pre-trained models during fine-tuning. To alleviate the gradient bias, our proposed method estimates unobservable gradients that minimize risk in unseen domains based on two key components: a task expert and a generalization expert. Experimental results on DOMAINBED show that our proposed method outperforms baseline methods in domain generalization. Through extensive analyses, we also demonstrate that the estimated unobservable gradients effectively reduce gradient bias, thereby helping to learn task-specific knowledge without hurting the generalization power of large-scale pre-trained models.

Although we verify the effectiveness of our proposed method, it heavily relies on the capability of pre-trained models. When unseen domains that pre-trained models did not encounter are given (*e.g.* ResNet trained on ImageNet does not see medical images), the pre-trained models might not act as an approximation of the oracle model of the domains. We will address this issue in future work.

## REPRODUCIBILITY STATEMENT

We provide the source code for reproduction in the supplementary materials. See Appendix A for the hyperparameters used for the experiments.

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

APPENDIX

## A   IMPLEMENTATION DETAILS

**Hyperparameter search strategy.**   Similar to  Cha et al. (2022), the hyperparameter tuning strategy differs depending on which pre-trained model is used. In the experiments of this work, we use three different pre-trained models: ResNet-50 (He et al., 2016) pre-trained on ImageNet (Deng et al., 2009) (`RN50`), ViT-B/16 (Dosovitskiy et al., 2021) with CLIP (Radford et al., 2021) (`CLIP`), and RegNetY-16GF (Radosavovic et al., 2020) with SWAG (Singh et al., 2022) (`SWAG`).

Table 6: Hyperparameters used for `RN50` in the experiments.

| Hyperparameter | PACS | VLCS | OfficeHome | TerraInc | DomainNet |
|---|---|---|---|---|---|
| $\lambda$ | 0.01 | 0.05 | 0.01 | 0.01 | 0.01 |
| Learning rate | 5e-5 | 5e-5 | 5e-5 | 5e-5 | 5e-5 |
| Weight decay | 0.0 | 1e-4 | 1e-6 | 0.0 | 1e-4 |
| Dropout | 0.0 | 0.5 | 0.5 | 0.0 | 0.1 |

A two-stage hyperparameter search strategy is used for experiments with `RN50`. Here, the batch size and the moving average coefficient ($m$) are fixed as 32 and 0.999 in the entire search procedure, respectively. In the first stage, we search the gradient scale factor ($\lambda$) from {0.01, 0.05, 0.1, 0.5}. In this stage, we fix the learning rate as 5e-5 and do not use weight decay and dropout (*i.e.,* weight decay and dropout are equal to 0). In the second stage, we fix $\lambda$ as the one searched in the first stage. Then, we search the learning rate from {1e-5, 3e-5, 5e-5}, weight decay from {0, 1e-6, 1e-4}, and dropout from {0, 0.1, 0.5}. We provide the hyperparameters we use for `RN50` in Table 6.

Table 7: $\lambda$ used for `CLIP` and `SWAG` in the experiments.

| Pre-trained Model | PACS | VLCS | OfficeHome | TerraInc | DomainNet |
|---|---|---|---|---|---|
| CLIP | 0.05 | 0.1 | 0.05 | 0.05 | 0.05 |
| SWAG | 0.05 | 0.1 | 0.05 | 0.05 | 0.05 |

Unlike the experiments with `RN50`, we apply single-stage hyperparameter search strategy to `CLIP` and `SWAG`. Here, we only search $\lambda$ from {0.01, 0.05, 0.1, 0.5} with hyperparameters such as the batch size, learning rate, weight decay, and dropout fixed. In particular, we fix the learning rate, weight decay, and dropout as the first stage of the hyperparameter search of `RN50`. For the batch size, we fix the batch size as 32 except for the experiments with DomainNet (Peng et al., 2019) where the batch size is fixed as 24 for the experiments with `CLIP`. For `SWAG`, we fix the batch size as 16 for all experiments. In Table 7, we show what $\lambda$ we use for `CLIP` and `SWAG`. Similar to Cha et al. (2022), we fix the number of iterations as 15,000 for DomainNet and 5,000 for the others regardless of pre-trained models.

## B  ADDITIONAL RESULTS

### B.1  MAIN RESULTS

Table 8: Domain generalization accuracy (%) on the five domain generalization benchmark datasets with the three different pre-trained models. We mark ∗, †, and ‡ for the results from Gulrajani & Lopez-Paz (2021), Cha et al. (2021) and Cha et al. (2022) respectively. We use the reported numbers from each paper for Fish, Fishr, SelfReg, mDSDI, GVRT, and SMA.

| Method | PACS | VLCS | OfficeHome | TerraInc | DomainNet | Avg. |
|---|---|---|---|---|---|---|
| *Using ResNet-50 pre-trained on ImageNet.* | | | | | | |
| MMD∗ | 84.7 ±0.5 | 77.5 ±0.9 | 66.3 ±0.1 | 42.2 ±1.6 | 23.4 ±9.5 | 58.8 |
| MixStyle† | 85.2 ±0.3 | 77.9 ±0.5 | 60.4 ±0.3 | 44.0 ±0.7 | 34.0 ±0.1 | 60.3 |
| GroupDRO∗ | 84.4 ±0.8 | 76.7 ±0.6 | 66.0 ±0.7 | 43.2 ±1.1 | 33.3 ±0.2 | 60.7 |
| IRM∗ | 83.5 ±0.8 | 78.5 ±0.5 | 64.3 ±2.2 | 47.6 ±0.8 | 33.9 ±2.8 | 61.6 |
| ARM∗ | 85.1 ±0.4 | 77.6 ±0.3 | 64.8 ±0.3 | 45.5 ±0.3 | 35.5 ±0.2 | 61.7 |
| VREx∗ | 84.9 ±0.6 | 78.3 ±0.2 | 66.4 ±0.6 | 46.4 ±0.6 | 33.6 ±2.9 | 61.9 |
| CDANN∗ | 82.6 ±0.9 | 77.5 ±0.1 | 65.8 ±1.3 | 45.8 ±1.6 | 38.3 ±0.3 | 62.0 |
| DANN∗ | 83.6 ±0.4 | 78.6 ±0.4 | 65.9 ±0.6 | 46.7 ±0.5 | 38.3 ±0.1 | 62.6 |
| RSC∗ | 85.2 ±0.9 | 77.1 ±0.5 | 65.5 ±0.9 | 46.6 ±1.0 | 38.9 ±0.5 | 62.7 |
| MTL∗ | 84.6 ±0.5 | 77.2 ±0.4 | 66.4 ±0.5 | 45.6 ±1.2 | 40.6 ±0.1 | 62.9 |
| Mixup∗ | 84.6 ±0.6 | 77.4 ±0.6 | 68.1 ±0.3 | 47.9 ±0.8 | 39.2 ±0.1 | 63.4 |
| MLDG∗ | 84.9 ±1.0 | 77.2 ±0.4 | 66.8 ±0.6 | 47.7 ±0.9 | 41.2 ±0.1 | 63.6 |
| Fish | 85.5 ±0.3 | 77.8 ±0.3 | 68.6 ±0.4 | 45.1 ±1.3 | 42.7 ±0.2 | 63.9 |
| Fishr | 85.5 ±0.4 | 77.8 ±0.1 | 67.8 ±0.1 | 47.4 ±1.6 | 41.7 ±0.0 | 64.0 |
| ERM† | 84.2 ±0.1 | 77.3 ±0.1 | 67.6 ±0.2 | 47.8 ±0.6 | 44.0 ±0.1 | 64.2 |
| SagNet∗ | 86.3 ±0.2 | 77.8 ±0.5 | 68.1 ±0.1 | 48.6 ±1.0 | 40.3 ±0.1 | 64.2 |
| SelfReg | 85.6 ±0.4 | 77.8 ±0.9 | 67.9 ±0.7 | 47.0 ±0.3 | 42.8 ±0.0 | 64.2 |
| CORAL∗ | 86.2 ±0.3 | 78.8 ±0.6 | 68.7 ±0.3 | 47.6 ±1.0 | 41.5 ±0.1 | 64.5 |
| mDSDI | 86.2 ±0.2 | 79.0 ±0.3 | 69.2 ±0.4 | 48.1 ±1.4 | 42.8 ±0.1 | 65.1 |
| GVRT | 85.1 ±0.3 | 79.0 ±0.2 | 70.1 ±0.1 | 48.0 ±1.4 | 44.1 ±0.1 | 65.2 |
| MIRO‡ | 85.4 ±0.4 | 79.0 ±0.0 | 70.5 ±0.4 | 50.4 ±1.1 | 44.3 ±0.2 | 65.9 |
| SMA | 87.5 ±0.2 | 78.2 ±0.2 | 70.6 ±0.1 | 50.3 ±0.5 | 46.0 ±0.1 | 66.5 |
| SWAD† | **88.1** ±0.1 | 79.1 ±0.1 | 70.6 ±0.2 | 50.0 ±0.3 | **46.5** ±0.1 | 66.9 |
| **GESTUR** | 88.0 ±0.2 | **80.1** ±0.2 | **71.1** ±0.0 | **51.3** ±0.2 | 46.3 ±0.1 | **67.4** |
| *Using ViT-B/16 with CLIP.* | | | | | | |
| ERM‡ | 83.4 ±0.5 | 75.9 ±1.3 | 66.4 ±0.5 | 35.3 ±0.8 | 44.4 ±0.6 | 61.1 |
| SWAD | 91.3 ±0.1 | 79.4 ±0.4 | 76.9 ±0.1 | 45.4 ±0.5 | 51.7 ±0.8 | 68.9 |
| MIRO‡ | 95.6 ±0.8 | 82.2 ±0.3 | 82.5 ±0.1 | 54.3 ±0.4 | 54.0 ±0.3 | 73.7 |
| **GESTUR** | **96.0** ±0.0 | **82.8** ±0.1 | **84.2** ±0.1 | **55.7** ±0.2 | **58.9** ±0.1 | **75.5** |
| *Using RegNetY-16GF with SWAG.* | | | | | | |
| ERM‡ | 89.6 ±0.4 | 78.6 ±0.3 | 71.9 ±0.6 | 51.4 ±1.8 | 48.5 ±0.6 | 68.0 |
| SWAD‡ | 94.7 ±0.2 | 79.7 ±0.2 | 80.0 ±0.1 | 57.9 ±0.7 | 53.6 ±0.6 | 73.2 |
| MIRO‡ | **97.4** ±0.2 | 79.9 ±0.6 | 80.4 ±0.2 | 58.9 ±1.3 | 53.8 ±0.1 | 74.1 |
| SMA | 95.5 ±0.0 | 80.7 ±0.1 | 82.0 ±0.0 | 59.7 ±0.0 | 60.0 ±0.0 | 75.6 |
| **GESTUR** | 96.9 ±0.1 | **83.5** ±0.1 | **83.1** ±0.0 | **61.1** ±0.4 | **60.1** ±0.0 | **76.9** |

In § 3.2, we only compare baselines superior to ERM (Vapnik, 1999) with GESTUR for simplicity. Here, we provide the entire results of the main experiment in Table 8.

**Baselines.**  In the main experiment, we compare GESTUR against a number of baselines: MMD (Li et al., 2018b), MixStyle (Zhou et al., 2021), GroupDRO (Sagawa et al., 2019), IRM (Arjovsky et al., 2019), ARM (Zhang et al., 2021b), VREx (Krueger et al., 2021), CDANN (Li et al., 2018c), DANN (Ganin et al., 2016), RSC (Huang et al., 2020), MTL (Blanchard et al., 2021), Mixup (Wang et al., 2020; Xu et al., 2020; Yan et al., 2020), MLDG (Li et al., 2018a), Fish (Shi et al., 2022), Fishr (Rame et al., 2022), ERM (Vapnik, 1999), SagNet (Nam et al., 2021), Self-

Reg (Kim et al., 2021), CORAL (Sun & Saenko, 2016), mDSDI (Bui et al., 2021), GVRT (Min et al., 2022), MIRO (Cha et al., 2022), SWAD (Cha et al., 2021), and SMA (Arpit et al., 2022).

## B.2 APPLICABILITY OF SWAD (CHA ET AL., 2021) TO GESTUR

Table 9: Evaluation results (%) of combination of SWAD and GESTUR on the four datasets with the three different pre-trained models.

| Method | PACS | VLCS | OfficeHome | TerraInc | Avg. |
|---|---|---|---|---|---|
| *Using ResNet-50 pre-trained on ImageNet.* | | | | | |
| GESTUR | 88.0 $\pm0.2$ | 80.1 $\pm0.2$ | 71.1 $\pm0.0$ | 51.3 $\pm0.2$ | 72.6 |
| GESTUR + SWAD | 88.3 $\pm0.1$ | 80.1 $\pm0.1$ | 71.0 $\pm0.0$ | 51.2 $\pm0.2$ | 72.7 |
| *Using ViT-B/16 with CLIP.* | | | | | |
| GESTUR | 96.0 $\pm0.0$ | 82.8 $\pm0.1$ | 84.2 $\pm0.1$ | 55.7 $\pm0.2$ | 79.7 |
| GESTUR + SWAD | 95.9 $\pm0.0$ | 82.8 $\pm0.1$ | 84.3 $\pm0.0$ | 55.3 $\pm0.6$ | 79.6 |
| *Using RegNetY-16GF with SWAG.* | | | | | |
| GESTUR | 96.9 $\pm0.1$ | 83.5 $\pm0.1$ | 83.1 $\pm0.0$ | 61.1 $\pm0.4$ | 81.2 |
| GESTUR + SWAD | 96.8 $\pm0.0$ | 83.0 $\pm0.1$ | 83.4 $\pm0.1$ | 60.6 $\pm0.8$ | 81.0 |

**Setup.** The recent study (Cha et al., 2022) has observed that SWAD (Cha et al., 2021) that seeks the flat minima is a good optimizer for domain generalization, improving the generalization performance of several baselines by applying it to the baselines as a optimizer. Motivated by this observation, we evaluate the performance of our GESTUR applied with SWAD as a optimizer to verify whether GESTUR and SWAD are orthogonal directions to each other.

**Results.** Table 9 shows that SWAD does not improve the performance of GESTUR. We conjecture that it is because EMA used to transfer the knowledge of TE to GE has a similar effect as SWAD to find a flat minima by averaging the model's weights.

## B.3 COMPARISON WITH CLIP-BASED BASELINES

Table 10: Evaluation results (%) on the four datasets with CLIP. Here, we compare GESTUR with CLIP-based baelines, CILP Zero-shot and WiSE-FT (Wortsman et al., 2022).

| Method | PACS | VLCS | OfficeHome | TerraInc | Avg. |
|---|---|---|---|---|---|
| CLIP Zero-shot | **96.8** $\pm0.0$ | 81.7 $\pm0.3$ | 83.0 $\pm0.3$ | 31.3 $\pm0.2$ | 73.2 |
| WiSE-FT ($\alpha = 0.5$) | 94.5 $\pm0.0$ | **83.9** $\pm0.3$ | 83.9 $\pm0.2$ | 47.5 $\pm1.2$ | 77.5 |
| **GESTUR** | 96.0 $\pm0.0$ | 82.8 $\pm0.1$ | **84.2** $\pm0.1$ | **55.7** $\pm0.2$ | **79.7** |

**Setup.** CLIP (Radford et al., 2021) is pre-trained on the huge web-crawled image-caption pair dataset and has been widely adopted in various computer vision tasks due to its generalization ability. CLIP-based methods could be strong baselines in domain generalization because the text content they used in pre-training could act as a robust anchor to the domain shift of images. Therefore, we conduct additional experiments using CLIP-based methods, CLIP Zero-shot and WiSE-FT (Wortsman et al., 2022). The CLIP-based methods require text-based queries to output text-based representations of target classes. Following the previous study, we obtain the 80 text-based queries from the official repository[1] of CLIP and compute the final text-based representation of each target class

---

[1] `https://github.com/openai/CLIP/blob/main/notebooks/Prompt_Engineering_for_ImageNet.ipynb`

by averaging the text-based representations of the queries. Finally, the model predictions are computed with the text-based representations and the representations of input images. For WiSE-FT, an ensemble of the fine-tuned and zero-shot models, we set the balance factor $\alpha$ as $0.5$ following its original paper since target unseen domains are inaccessible in the domain generalization setting.

**Results.** Table 10 shows the evaluation results where GESTUR achieves the best averaged performance. In detail, GESTUR outperforms CLIP Zero-shot on VLCS, OfficeHome, and TerraInc, and shows comparable performance on PACS. Likewise, GESTUR achieves better performance on PACS, OfficeHome, and TerraInc than WiSE-FT and comparable performance on VLCS.

Interestingly, the CLIP-based methods exhibit severe performance degradation on TerraInc. We conjecture that their performance is sensitive to pre-defined text-based queries. For example, the query "a sketch of a {}" is helpful for the "Sketch" domain of PACS. On the other hand, the queries "a plastic {}" and "a {} in a video game" are not helpful for TerraInc, which is composed of animal images taken from the wild. These observations indicate that the CLIP-based methods require hard prompt engineering for each target dataset. Moreover, the CLIP-based methods solely depend on `CLIP`, which cannot be extended to other architecture or learning methods trained on only visual modality, such as ResNet with ImageNet and RegNet with SWAG. Considering these, our GESTUR achieves a meaningful performance.

# C FURTHER ANALYSIS

## C.1 RELATIONSHIP BETWEEN $\lambda$ AND THE TYPES OF THE PRE-TRAINED MODEL

Table 11: Evaluation results (%) on VLCS with the three different pre-trained models varying $\lambda$.

| Dataset (size) | Pre-training | Architecture | $\lambda$ | | | |
|---|---|---|---|---|---|---|
| | | | 0.01 | 0.05 | 0.1 | 0.5 |
| ImageNet (1.3M) | ERM | ResNet-50 | 78.9 ±0.3 | **80.1** ±0.2 | 80.0 ±0.1 | 77.6 ±0.1 |
| CLIP (400M) | CLIP | ViT-B/16 | 81.3 ±0.4 | 82.7 ±0.1 | **82.8** ±0.1 | 82.1 ±0.3 |
| Instagram (3.6B) | SWAG | RegNetY-16GF | 81.7 ±0.0 | 82.7 ±0.2 | **83.5** ±0.1 | 82.4 ±0.2 |

Table 12: Evaluation results (%) on OfficeHome with the three different pre-trained models varying $\lambda$.

| Dataset (size) | Pre-training | Architecture | $\lambda$ | | | |
|---|---|---|---|---|---|---|
| | | | 0.01 | 0.05 | 0.1 | 0.5 |
| ImageNet (1.3M) | ERM | ResNet-50 | **71.1** ±0.0 | **71.1** ±0.1 | 70.4 ±0.2 | 68.9 ±0.1 |
| CLIP (400M) | CLIP | ViT-B/16 | 82.5 ±0.2 | 84.2 ±0.1 | 84.4 ±0.0 | **84.7** ±0.0 |
| Instagram (3.6B) | SWAG | RegNetY-16GF | 81.5 ±0.2 | 83.1 ±0.0 | **83.5** ±0.0 | 81.1 ±0.1 |

Table 13: Evaluation results (%) on TerraIncognita with the three different pre-trained models varying $\lambda$.

| Dataset (size) | Pre-training | Architecture | $\lambda$ | | | |
|---|---|---|---|---|---|---|
| | | | 0.01 | 0.05 | 0.1 | 0.5 |
| ImageNet (1.3M) | ERM | ResNet-50 | **51.3** ±0.2 | 50.0 ±0.4 | 45.5 ±0.2 | 31.2 ±0.1 |
| CLIP (400M) | CLIP | ViT-B/16 | 51.3 ±0.2 | **55.7** ±0.2 | 54.0 ±0.3 | 42.3 ±0.9 |
| Instagram (3.6B) | SWAG | RegNetY-16GF | 57.6 ±0.9 | 61.1 ±0.4 | **62.1** ±0.3 | 54.9 ±0.1 |

In § 3.6, we analyze the relationship between $\lambda$ and the size of the pre-trained model. However, we only present the results from PACS (Li et al., 2017) in Table 5 for simplicity. Here, we provide the additional results from VLCS (Fang et al., 2013), OfficeHome (Venkateswara et al., 2017), and TerraIncognita (Beery et al., 2018) in Table 11, Table 12, and Table 13, respectively.

## C.2 PERFORMANCE ON SOURCE DOMAINS

Table 14: Evaluation results (%) on the four datasets with RN50. Here, we average the performances in the source domains, not the performance in the unseen target domain.

| Method | PACS | VLCS | OfficeHome | TerraInc | Avg. |
|---|---|---|---|---|---|
| ERM | 97.4 ±0.2 | 86.7 ±0.1 | 82.9 ±0.3 | **92.2** ±0.1 | 89.8 |
| GESTUR w/ TE | 97.1 ±0.1 | 86.9 ±0.2 | 81.7 ±0.2 | 89.4 ±0.1 | 88.8 |
| **GESTUR w/ GE** | **98.2** ±0.1 | **87.4** ±0.2 | **84.8** ±0.3 | 91.4 ±0.1 | **90.5** |

**Setup.** Domain generalization aims to improve the generalization performance on unseen domains shifted from source domains. Thus, domain generalization studies often do not consider situations where the source domains and the target domains are similar. To verify whether estimated unobservable gradients are useful when the unseen domains are similar to the source domains, we report

the performance on the training-domain validation set, simulating the situations when the testing domains are exactly the same as the training domains.

**Results.** The evaluation results are summarized in Table 14. GESTUR w/ TE shows worse performance than ERM, indicating that the estimated unobservable gradients act as noisy gradients. Namely, gradients biased toward the source domains are more helpful than estimated unobservable gradients when the source domains and the target domains are similar. Nonetheless, GESTUR w/ GE performs better than ERM, demonstrating that our two-expert architecture is robust to various situations even when source domains and unseen domains are similar or not.

### C.3 SIMILARITY BETWEEN TRUE UNOBSERVABLE GRADIENTS $\mathbf{g}_u$ AND ESTIMATED UNOBSERVABLE GRADIENTS $\tilde{\mathbf{g}}_u$ OF GESTUR

**Setup.** In this paper, we argue that gradient bias is a major culprit in degrading domain generalization performance (Figure 1a) and our proposed method relieves the gradient bias by estimating unobservable gradients. To support this argument, we reported the number of iterations where gradient conflicts exist in Figure 1b and Table 3. To examine whether the estimated unobservable gradients $\tilde{\mathbf{g}}_u$ are similar to the true unobservable gradients $\mathbf{g}_u$, we add the analysis calculating the cosine similarity of the true and estimated unobservable gradients. Note that the true unobservable gradients are computed by cross-entropy loss using true labels of unseen domain datasets $\mathcal{D}_u$. On the other hand, the estimated unobservable gradients are just computed as the parameter difference between GE and TE ($\theta_{GE} - \theta_{TE}$).

**Results.** Figure 2 shows that our estimated gradients display positive similarity scores with the true gradients. This trend demonstrates that the estimated gradients reduce the number of gradient conflicts, leading models to reduce the risks of unseen domains without accessing unseen domain data.

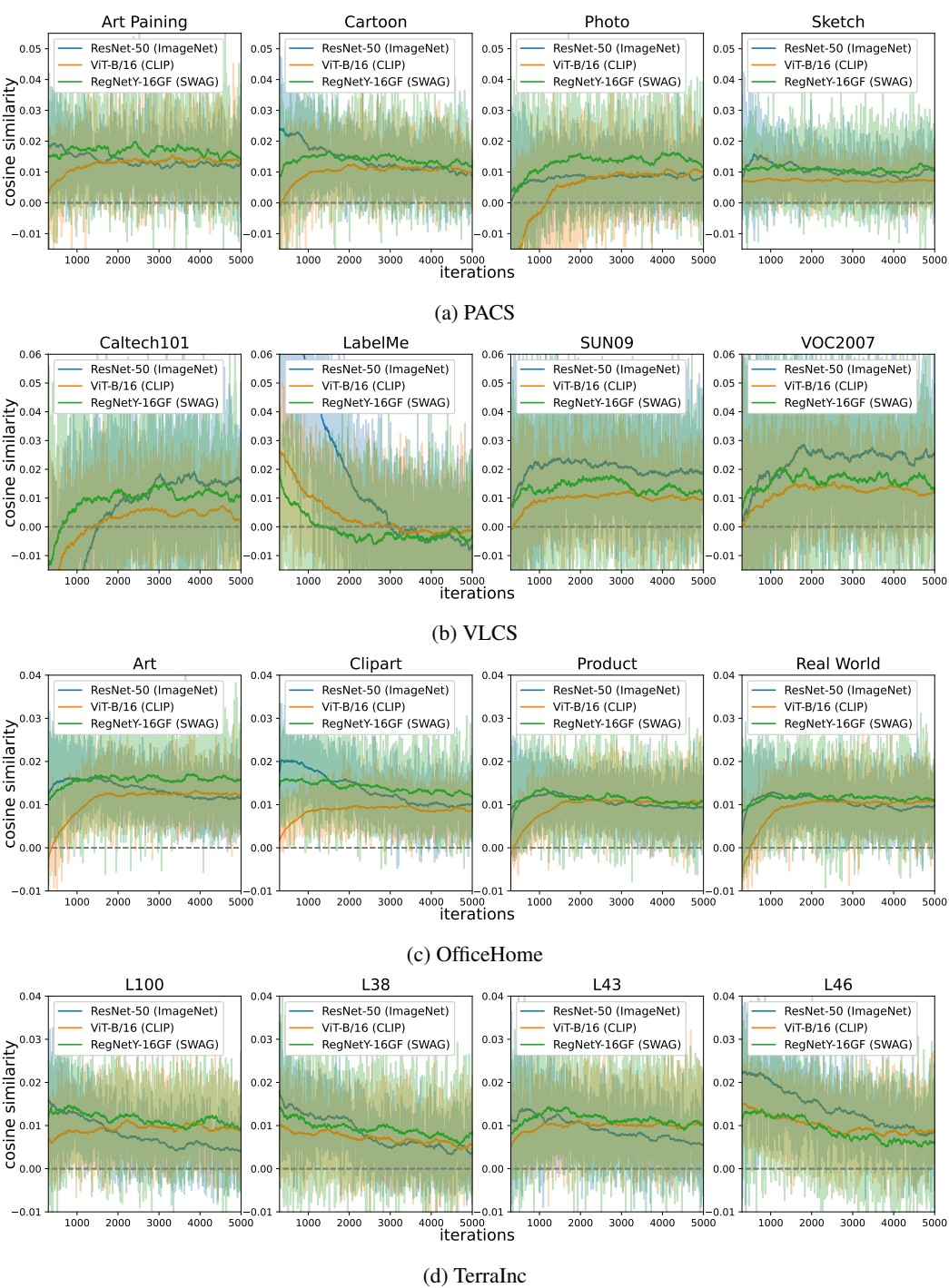

Figure 2: Cosine similarity between the true unobservable gradients $\mathbf{g}_u$ and the estimated unobservable gradients $\tilde{\mathbf{g}}_u$

