# OpenReview forum: "Gradient Estimation for Unseen Domain Risk Minimization with Pre-Trained Models"
_ICLR.cc/2023/Conference — Submitted to ICLR 2023_

### Official Review · Reviewer_e3A6 · 2022-10-24

**Confidence:** 4
**Clarity, Quality, Novelty And Reproducibility:** 1. Clarity
**Correctness:** 3
**Technical Novelty And Significance:** 3
**Empirical Novelty And Significance:** 3
**Recommendation:** 5

**Strength And Weaknesses:**

### Pros
1. The proposed method GESTUR is simple and well-motivated.

2. GESTUR is competitive on various domain generalization benchmarks, verified by empirical results.

3. The analysis and ablation are comprehensive, well justifying the proposed method.

### Cons
1. The hyperparameter $\lambda$ is very sensitive to domain generalization tasks, as verified by Table 5 and B.2 in the appendix. I am still confused about how to choose $\lambda$? What does "while those of the unseen domain are used for hyperparameter search." in Section 3.1 mean? DomainBed provides three model/hyperparameter selection protocols, which protocol does this paper adopt, source validation set, oracle test-domain validation set, or leave-one-domain-out cross-validation?

2. In MIRO paper, MIRO can combine with SWAD to further improve the generalization performance, average accuracy of 68.1 for the ResNet-50 model. How about GESTUR in this paper?


**Summary Of The Paper:**

This paper studies the problem of domain generalization. The authors propose to reduce the gradient conflict between the gradient biased toward the source domains and the unobservable gradient that could minimize risks in unseen domains. Specifically, the authors leverage the large-scale pre-trained model as a loose approximation of the oracle model for the unseen domains and propose a simple yet effective method called GESTUR. In GESTUR, a fast-update model called task expert (TE) is directly optimized on source domains with calibrated gradient. A slow-update model called generalization expert (GE) is EMA updated by TE and provides the approximated unobservable gradient for calibration. Empirical results on five popular domain generalization benchmarks show that GESTUR outperforms other methods.

**Summary Of The Review:**

This paper proposes a simple yet effective method for domain generalization. Extensive experiments well justify the advantage of the proposed method. My main concern is how to select the sensitive hyperparameter $\lambda$ without accessing unseen target domain data.

***********Post-rebuttal**************
After reading all the discussions and reviews, I share the same concern on the novelty of this submission as Reviewer seAG and Reviewer TqRb. Also, I also agree with Reviewer ENaU that the feasibility and reliability of using large-scale pre-trained models as a loose approximate of unseen domains are not provable, limiting the proposed method. Therefore, I decided to decrease my rating to 5.

---

> ### Author Response · Authors · 2022-11-12
> **Author responses for Reviewer e3A6**
>
> We sincerely thank Reviewer *e3A6* for their constructive feedback again. We will address Reviewer *e3A6*'s concerns below.
>
> ---
>
> **About the hyperparameter search process:** While preparing an answer to Reviewer *e3A6*'s question about the hyperparameter search process, we found that we misunderstood the search process and, accordingly, reported the results of the main experiment incorrectly. We described that we found $\lambda$ in the *testing-domain validation set*, and reported the performance when using the hyperparameter found in this way, leading to misreading. However, it is correct that all our experiments should follow the *training-domain validation*, and it is also correct to report the performance using $\lambda$ searched on the *training-domain validation set*. Therefore, we update our proposed method’s performance in Figure 1b, Tables 1-8, and 11-13. Once again, we deeply apologize for the confusion.
>
> **About the applicability of *SWAD* to *GESTUR*:** This is a good question! We already tried to apply *SWAD* to *GESTUR* and observed the performance change. The evaluation results are added in Table 9 (Appendix B.2). As shown in the table, *SWAD* does not improve the performance of *GESTUR*. We conjecture that it is because EMA used to transfer the knowledge of TE to GE has a similar effect as *SWAD* to find a flat minima by averaging the model’s weights.

---

### Official Review · Reviewer_TqRb · 2022-10-24

**Confidence:** 4
**Correctness:** 3
**Technical Novelty And Significance:** 2
**Empirical Novelty And Significance:** 2
**Recommendation:** 5

**Clarity, Quality, Novelty And Reproducibility:**

Clarity: the presentation is generally clear in most parts of the paper.

Quality: fair.

Novelty: a bit low.

Reproducibility: it could be easy to reproduce due to the technical simplicity.

**Strength And Weaknesses:**

[+] The paper propose to use EMA to simultaneously learn the task-specific knowledge and preserving the generalization ability of large-scale pretrained model for domain generalization, which is interesting.

[+] The ablation studies and comparisons on several domain generalization datasets  support the effectiveness of the method.

[-] The paper uses pretrained models as the loose approximation of the oracle model of the unseen domains and and estimates the unobserved gradient of unseen domains. But it is not clear why the pretrained models is able to approximate the oracle model. For example, how can an ImageNet-pretrained ResNet can be the oracle model for unseen domains like “sketch”? More explanations and discussions are needed for this motivation.

[-] The reviewer is also concerned about the novelty of the proposed method. The estimated gradient in eq. 4 looks more like a regularization that protects the model from changing too much compared with the originally pretrained parameters, which may hurt its generalization ability. Eq. 6 also seems to be doing the same thing. However, this kind of techniques have already been explored in some continue learning methods to deal with catastrophic forgetting.


**Summary Of The Paper:**

The paper explores domain generalization by estimating gradients of unobserved domains using large-scale pretrained models. The method learns task-specific knowledge for the pretrained model while preserving its generalization ability with the estimated gradients by EMA. Results on several benchmarks on DomainBed show the effectiveness of the proposed method.

**Summary Of The Review:**

The paper proposes to fine-tune the large-scale pretrained model for domain generalization by learning task-specific knowledge from source domains while keeping the generalization ability of the pretrained models. But the motivation and justification behind the method are not clear and well supported by experiments. The technical novelty of the method seems weak to me. Therefore, I think it is below the acceptance threshold.

---

> ### Author Response · Authors · 2022-11-12
> **Author responses for Reviewer TqRb**
>
> We sincerely thank Reviewer *TqRb* for their constructive feedback again. We will address Reviewer *TqRb*'s concerns below.
>
> ---
>
> **About whether the pre-trained models can be an approximation of the oracle model in unseen domains:** Please refer to *General Response 2.*
>
>
> **About the novelty of our proposed method:** The novelty of our method is derived from the fact that we point out the gradients biased to only source domains as the cause of degradation of domain generalization. And, we show that unobservable gradients that minimize the risk of unseen domains could be estimated with large-scale pre-trained models, and it relieves the gradient bias.
>
> As Reviewer *TqRb* mentioned, our proposed method (especially Equation 4) shares a similar equation form with the regularization methods in related research fields, which prevent rapid target model changes using pre-trained models. For example, the regularizer term of *L$^2$-SP* (Xuhong, 2018) [3] looks similar to the term of our unobservable gradients $\tilde{\mathbf{g}}_u$:
>
> $L^2-SP: \mathbf{g} = 2\cdot\lambda(\theta - \theta^0)$, Ours: $\tilde{\mathbf{g}}_u = \lambda||\mathbf{g}||_2\cdot\frac{(\theta_\texttt{TE} - \theta_\texttt{GE})}{||(\theta_\texttt{TE} - \theta_\texttt{GE})||_2}$
>
> However, we argue that there is a difference in the motivation of the unobservable gradients which are estimated based on the assumption that large-scale pre-trained models could act as the approximation of the oracle model $\theta^*$. Moreover, since we interpret the term from the perspective of gradients, we adaptively adjust the scale of the estimated gradients considering the norm of the observable gradients ($||\mathbf{g}||_2\cdot\frac{1}{||(\theta_\texttt{TE} - \theta_\texttt{GE})||_2}$). In last, unlike the existing regularization term that freezes pre-trained models as $\theta^0$, we update the parameters of pre-trained models to additionally learn task-specific knowledge (Equation 6) because the pre-trained model is the loose approximation of the oracle model that does not equip the knowledge yet. It can be seen that the update of the pre-trained models is important in that our method outperforms *MIRO*, which performs better than *L$^2$-SP* in a recent study (Cha et al., 2022).
>
> ---
> **References**
>
> [3] Explicit inductive bias for transfer learning with convolutional networks, Xuhong, ICML (2018)

---

### Official Review · Reviewer_ENaU · 2022-10-25

**Confidence:** 3
**Correctness:** 3
**Technical Novelty And Significance:** 3
**Empirical Novelty And Significance:** 3
**Recommendation:** 5

**Clarity, Quality, Novelty And Reproducibility:**

The idea is novel, and using a pre-trained model to help the domain generalization is interesting. The paper is clear.


**Strength And Weaknesses:**

This paper proposes to estimate the unobservable gradients from a pre-trained model, which helps the model generalization across various domains. Results on various configurations show the ability of the proposed method.

There are some concerns with the paper:
1. The main idea of the paper is to reduce the influence from the source domain gradients and make the model generalize better. However, since we do not know which is the target domain, does it requires some kind of prior knowledge encoded in the pre-trained model? For example, if the target domain is very similar to (or distant from) the source domain, the pre-trained model will produce the same unobservable gradients, but not both of them help the generalization.
2. Do the experiments in Figure 1 depend on the relationship between the source and target domain? Could the authors show the similarity between the true gradient to the target domain and the gradient estimated by the pre-trained model?
3. Based on the results in Table 3.6 and supplementary, the value of $\lambda$ depends on the size of the pre-trained model. Does it relate to the similarity between the source and target domains? Or the diversity of the dataset of the pre-trained model?
4. Please consider more comparison methods in the experiments.

**Summary Of The Paper:**

This paper targets the domain generalization task. Based on the observation that fine-tuning may introduce gradient bias and hurt generalization ability, the paper estimates unobservable gradients that reduce potential risks in unseen domains. The main requirement is that there is a pre-trained model. Experimental results show that the proposed method outperforms baseline methods on DomainBed.


**Summary Of The Review:**

The overall quality of the paper is good, and the idea is interesting. The two main concerns of the paper are:
1. Theoretical or empirical analysis should be provided to support the motivation of the paper;
2. How to design a fair comparison with other methods since a pre-trained model is included.

---

> ### Author Response · Authors · 2022-11-12
> **Author responses for Reviewer ENaU**
>
> We sincerely thank Reviewer *ENaU* for their constructive feedback again. We will address Reviewer *ENaU*'s concerns below.
>
> ---
>
> **About whether the pre-trained models can be an approximation of the oracle model in unseen domains:** Please refer to *General Response 2.*
>
> **About the similarity of source domains $\mathcal{D}$ and unseen domains $\mathcal{D}_u$:** This is a good question! Domain generalization aims to improve the generalization performance on unseen domains shifted from source domains. Thus, domain generalization studies often do not consider situations where the source domains and the target domains are similar. To verify whether estimated unobservable gradients are useful when the unseen domains are similar to the source domains, we report the performance on the training-domain validation set, simulating the situations when the testing domains are exactly the same as the training domains. The evaluation results are summarized in Table 14 (Appendix C.2). As Reviewer *ENaU* pointed out, *GESTUR w/ TE* shows worse performance than ERM, indicating that the estimated unobservable gradients act as noisy gradients. Namely, gradients biased toward the source domains are more helpful than estimated unobservable gradients when the source domains and the target domains are similar. Nonetheless, *GESTUR w/ GE* performs better than ERM, demonstrating that our two-expert architecture is robust to various situations even when source domains and unseen domains are similar or not.
>
> **About the similarity of true unobservable gradients $\mathbf{g}_u$ and estimated unobservable gradients $\tilde{\mathbf{g}}_u$:** This is also a good question! In this paper, we argue that gradient bias is a major culprit in degrading domain generalization performance and our proposed method relieves the gradient bias by estimating unobservable gradients. To support this argument, we reported the number of iterations where gradient conflicts exist in Figure 1 (b) and Table 3. Reviewer *ENaU*'s comment gratefully suggested examining whether the estimated unobservable gradients are similar to the true unobservable gradients. Motivated by this, we add the analysis to Figure 2 (Appendix C.3). For all the pre-trained models, we calculate the cosine similarity of the true and estimated unobservable gradients. Note that the true unobservable gradients are computed by cross-entropy loss using true labels of unseen domain datasets. On the other hand, the estimated unobservable gradients are just computed as the parameter difference between GE and TE ($\theta_{GE} - \theta_{TE})$. As shown in the figure, our estimated gradients display positive similarity scores with the true gradients. This trend demonstrates that the estimated gradients reduce the number of gradient conflicts, leading models to reduce the risks of unseen domains without accessing unseen domain data.
>
> **About factors affecting the optimal $\lambda$:** This is a great question! In the initial version of our paper, we concluded that the optimal $\lambda$ is affected by the size of pre-trained models. As Reviewer *ENaU*’s comment, there may be other factors that may influence optimal $\lambda$ value. However, it is difficult to delicately examine which factor significantly influences the optimal $\lambda$ value because many variables are entangled affecting the performance.
>
> First, it is difficult to quantify the similarities between source and unseen domains. Indirectly, we tried to analyze the evaluation results concerning each target unseen domain qualitatively (e.g., (P,A,C) training -> (S) testing in the PACS dataset), but we failed to find a meaningful trend of optimal $\lambda$. Second, we further conducted controlled experiments fixing the pre-training method (*RotNet* (Gidaris et al., 2018) [2]) and architecture (ResNet-50) and changing the size of pre-training datasets (ImageNet1K & 22K) to verify whether the size of pre-training datasets affects the trend. But, we also failed to find any special trends according to the size of the pre-training datasets.
>
> Instead, we conjecture again that the trend of optimal $\lambda$ is mainly affected by the prior knowledge in pre-trained models for target unseen domains. We want to highlight that analyzing what affects prior knowledge is also difficult because it is intertwined with various factors such as the diversity and size of datasets, pre-training strategy, and the architecture of pre-trained models. Even we don't know what the target unseen domain will be. It would be another promising research topic. We clarified Section 3.6 reflecting these findings.
>
> **About the applicability of CLIP-based models as baselines**: Please refer to *General Response 1.*
>
> ---
> **References**
>
> [2] Unsupervised Representation Learning by Predicting Image Rotations, Gidaris et al., ICLR (2018)

---

> > ### Author Response · Authors · 2022-11-12
> > **We attach the further response**
> >
> > We attach the further response
> >
> > **About fair comparison**: All the methods including our method and baselines use a pre-trained model as a backbone model by initializing the feature extractor $\theta^{f}$ with the pre-trained model. Note that our GESTUR more effectively utilize the pre-trained model by assuming it as the approximation of the oracle model $\theta^*$ of unseen domains $\mathcal{D}_u$.

---

### Official Review · Reviewer_seAG · 2022-10-26

**Confidence:** 3
**Correctness:** 3
**Technical Novelty And Significance:** 3
**Empirical Novelty And Significance:** 3
**Recommendation:** 5

**Clarity, Quality, Novelty And Reproducibility:**

- Clarity: The paper is overall clearly written.

- Quality and Novelty: The proposed method is well-motivated and experimental results are promising. However, the method design shares a lot of similarity to a recent work [1], which receives little discussion in the paper. The paper quality can be much improved by carefully discussing its comparisons to related work.

- Reproducibility: The authors provide code to reproduce the experimental results.

**Strength And Weaknesses:**

Strengths:
- The proposed method is simple but shows promising results
- The key idea is well-motivated: large pre-trained models have strong generalization ability and fine-tuning impairs this characteristic; thus, one can do fine-tuning more conservatively to retain the models generalization ability.

Weaknesses:
- One main concern I have about the paper is that the proposed method is very related to a recent work [1], that proposed to linearly interpolate a fine-tuned model’s weight with the un-tuned model’s weight. The only difference lies in that [1] does the interpolation “after” the model is fine-tuned, while the proposed method in this paper does the interpolation “during” fine-tuning (in the optimization steps). [1] is only very briefly mentioned in the related work section. However, I believe properly comparing to [1] in the paper on both the method design as well as experimental results is needed to make the contribution of this work stronger.
- In the experiments, the evaluation metric only measures the model performance on the unseen domain. However, it’s generally also important to see the model performance on the source domain. It will give a more complete picture on the pros/cons of different methods. For example, maybe the simple ERM can achieve best performance on source domain by trading-off only a bit of its performance on target domain; while methods designed to optimize for target domain may trade-off more performance on source domain.
- I appreciate that the authors have conducted the analysis in Sec 3.4 to validate the original hypothesis of biased gradient. However, more details on the experimental setup are needed. Specifically, how are the gradients on the unseen domain calculated? Is it by using the actual $(x, y)$ pairs in the unseen domain?
- Another relevant baseline to consider in the experiments is the zero-shot model baseline. Particularly, with models like CLIP, one can directly do zero-shot predictions on the unseen domain. Will such a model without any fine-tuning achieve even better performance compared to the proposed method?
- Paper presentation can be improved. There are many repetitive texts in the Introduction section and the Method section. For example, the paragraph on “Task expert and generalization expert” in the Method section is repeated in the Introduction. Authors could use the space to provide more insights to the method design.

[1] Robust fine-tuning of zero-shot models. Wortsman et al. 2022.


**Summary Of The Paper:**

The paper proposes a simple method that leverages the generalization ability of large pre-trained models for domain generalization. The key idea is to use large pre-trained models to estimate the unobserved gradient that minimizes the risks in the target domain, mitigating the gradient biased towards to the source domain during the fine-tuning stage. Experimental results show that the approach is promising despite its simplicity.

**Summary Of The Review:**

Overall, the motivation and the method design is interesting. However, the paper currently lacks proper comparisons to related work, making the contribution less strong as it stands.

---

> ### Author Response · Authors · 2022-11-12
> **Author responses for Reviewer seAG**
>
> We sincerely thank Reviewer *seAG* for their constructive feedback again. We will address Reviewer *seAG*'s concerns below.
>
> ---
>
> **About the difference between our study and recent work (*WiSE-FT*)**: This is a good question! Their methods may seem similar to our method in that they interpolate fine-tuned and pre-trained models. However, there is a significant difference in the motivation of interpolation between both methods. *WiSE-FT* ensembles the fine-tuned model with a frozen pre-trained model to improve the robustness of the fine-tuned model (i.e., leveraging the pre-trained model for the fine-tuned model). On the other hand, *GESTUR* employs EMA to transfer task-specific knowledge of the fine-tuned model (TE) to the pre-trained model (GE) (i.e., leveraging the fine-tuned model for the pre-trained model). This is based on the assumption that the pre-trained model has a good generalization ability but not task-specific knowledge yet. Therefore, GE of *GESTUR* is updated by the interpolation during fine-tuning, unlike the frozen pre-trained model of *WiSE-FT*. By doing so, the fine-tuned model and the pre-trained model are communicated while they are trained. We also add the description referring to the difference between GESTUR and WiSE-FT to Related Work Section.
>
> We want to emphasize that our technical novelty is not limited to the interpolation (EMA) between the fine-tuned model (TE) and pre-trained model (GE). Our technical novelty is also derived from the proposed fine-tuning strategy that improves the generalization ability of the fine-tuned model with the relieved gradients by estimating unobservable gradients while the model learns task-specific knowledge.
>
> **About the importance of performance on source domains**: We absolutely agree with Reviewer seAG’s comment about the performance on source domains, which is commonly overlooked in domain generalization studies. A good domain generalization method should also perform well in source domains. We report the source domain performance of *ERM* and *GESTUR* in Table 14 (Appendix C.2). *GESTUR* also achieves better or comparable performance on source domains, indicating that our two-experts architecture is also helpful for source domains.
>
> **About the way of calculating gradient conflicts**: As described in Section 3.4, we directly use the target unseen domain dataset $\mathcal{D}_u$. For every iteration, we first sample two mini-batches $\mathcal{B}\sim\mathcal{D}$, $\mathcal{B}_u \sim \mathcal{D}_u$ from source domains and unseen domains, respectively. And, we compute the losses of the mini-batches based on the current task-expert model $\theta_\texttt{TE}$, then calculate two gradients $\mathbf{g}, \mathbf{g}_u$ from the losses, respectively. Finally, we count the number of gradient conflicts ($\mathbf{g}\cdot\mathbf{g}_u < 0$) throughout the entire interaction. For GESTUR, we use the relieved gradient by the estimated gradient $\hat{\mathbf{g}}_u$. Note that the task-expert is updated by only the gradient $\mathbf{g}$ of the source domains because the unseen domains are inaccessible in fact.
>
> **About the applicability of CLIP-based models as baselines**: Please refer to *General Response 1.*

---

> ### Comment · Reviewer_seAG · 2022-11-22
> **Thank you for the response**
>
> Thank you to the authors for providing the responses. I agree that the motivation in this work is quite different to WiSE-FT. However, I still believe that the final methods derived are quite similar, and it should be discussed more carefully and compared experimentally. I believe the comparison could make the paper stronger. I would remain my original score.

---

> > ### Author Response · Authors · 2022-11-22
> > **We attach the additional experimental comparison with WiSE-FT**
> >
> > Thank you for your response.
> >
> > In response to your first review, we conducted further comparison experiments with WiSE-FT, which are in Appendix B.3. Also, the difference with WiSE-FT was reflected in Related Work ('*Although GESTUR’s EMA looks similar to their interpolation, GESTUR updates the pre-trained model to inject task-specific knowledge*'). We will attach that comparison table below. Note that CLIP Zero-shot and WiSE-FT require text-based queries which contain external powerful knowledge about target classes (it seems unfair for our method). Thus, we reported this table in Appendix rather than the main experimental tables.
> >
> > Method | PACS | VLCS | OfficeHome | TerraInc | Avg.
> > ----------|--------|--------|---------------|-----------|---------
> > CLIP Zero-shot | **96.8±0.0** | 81.7±0.3 | 83.0±0.3 | 31.3±0.2 |73.2
> > WiSE-FT (α = 0.5) | 94.5±0.0 | **83.9±0.3** | 83.9±0.2|47.5±1.2|77.5
> > GESTUR | 96.0±0.0|82.8±0.1|**84.2±0.1**|**55.7±0.2**| **79.7**
> >
> > Please refer to our general response for the additional experimental details.

---

### Author Response · Authors · 2022-11-12
**General responses for reviewers**

We sincerely thank all the reviewers for their constructive feedback. We are especially pleased that reviewers agree our proposed method (GESTUR) is well-motivated (*seAG*, *e3A6*), interesting (*seAG*, *ENaU*, *TqRb*), novel (*seAG*, *ENaU*, *e3A6*), and simple yet effective (*seAG*, *e3A6*). We are also glad that reviewers think our paper is well-written (*seAG*, *ENaU*, *e3A6*), our contributions are significant (*seAG*, *e3A6*, *ENaU*), and experiments and analysis are conducted well (*seAG*, *TqRb*, *e3A6*). During the rebuttal stage, we will try our best to address reviewers' concerns and make our contributions clear from invaluable feedback.

---
### General Response

We recap the goal of this study briefly:

**What is our goal?** (1) Demonstrate that computed gradients are biased toward only source domains, hurting the generalization performance on unseen domains. (2) Show that the biased gradients could be relieved by estimating unobservable gradients that minimize risks of unseen domains, based on the previously adopted assumption that large-scale pre-trained models approximate the oracle model of the unseen domains. (3) Show that the large-scale pre-trained models could get closer to the oracle model by injecting task-specific knowledge delicately. (4) Propose a novel domain generalization method that estimates unobservable gradients of unseen domains, relieve the biased gradients, and transfers the learned task-specific knowledge to the large-scale pre-trained model.

**What is *not* our goal?** We do not aim to prove that large-scale pre-trained models could act as the approximation of the oracle model of unseen domains.

---

> ### Author Response · Authors · 2022-11-12
> **General responses for reviewers**
>
>
> 1. **About more baselines (Reviewer *seAG*,*ENaU*)**
>
>     We exhaustively compared our method with 23 baseline methods, including recently proposed methods (*MixStyle* (2021), *ARM* (2021), *VREx* (2021), *MTL* (2021), *SagNet* (2021), *SelfReg* (2021), *mDSDI* (2021), *SWAD* (2021), *Fish* (2022), *Fishr* (2022), *GVRT* (2022), SMA (2022) and *MIRO* (2022)), and reported the entire evaluation results in Table 8 (Appendix B.1) and the short version in Table 1. We newly add SMA [1].
>
>     As suggested by Reviewer *seAG*, we also agree that CLIP-based methods (e.g., *WiSE-FT* (Wortsman et al., 2022)) could be strong baselines because the text content they used in pre-training could act as a robust anchor to the domain shift of visual content. Therefore, we conduct additional experiments using CLIP-based methods, but we consider that directly comparing our proposed method with the CLIP-based methods is somewhat inappropriate for the following reasons:
>
>     * (1) Such CLIP-based methods solely depend on multi-modal pre-trained models (i.e., CLIP), which cannot be extended to other architecture or learning methods trained on only visual modality, such as ResNet with ImageNet and RegNet with SWAG.
>     * (2) Such CLIP-based methods utilize text-based queries, which contain powerful external knowledge about target classes and require manual prompt engineering.
>
>     Thus, we add the additional analysis section comparing our proposed method with the CLIP-based methods, and report the evaluation results in Table 10 (Appendix B.3) rather than the main table (Table 1). As shown in the table, our proposed *GESTUR* outperforms the CLIP-based methods except for two cases where *CLIP zero-shot* and *WiSE-FT* perform better on PACS and VLCS, respectively. Note that these CLIP-based methods further utilize text-based queries which provide powerful external knowledge for the target classes.
>
>
> 2. **About whether the pre-trained models can be an approximation of the oracle model in unseen domains (Reviewer *ENaU*,*TqRb***)
>
>     It is the most important assumption of our proposed method that large-scale pre-trained models are the approximation of the oracle model in unseen domains (in this paper, we make this assumption more loosened). This assumption is first adopted in a recent study (Cha et al., 2022), and is empirically supported that it is helpful for generalization performance. The reason is that large-scale pre-trained models are highly likely to meet data from various domains during pre-training and encode knowledge of the domains.
>
>     We agree that pre-trained models cannot act as the approximation of the oracle model when pre-trained models do not encounter data of target unseen domains. It is the limitation of our proposed method, which is already described in Conclusion and Future Work Section (Section 5). Conversely, our method further improved the model performance in CLIP and SWAG (Table 1), which were pre-trained on large-scale web-crawled data and Instagram images from various domains, respectively. It is because that CLIP and SWAG were highly likely to meet data of target unseen domains during pre-training, leading them to act as the better approximation of the oracle model, making it possible to estimate the unobservable gradients more accurately.
>
>     We want to emphasize again that the scope of this paper is not to prove that pre-trained models can play the role of the oracle model, but show that unobservable gradients of unseen domains can be estimated with a large-scale pre-trained model. We believe that our method can contribute more to improving domain generation performance when large-scale pre-trained models with excellent generalization ability are developed in the future.
>
>
> 3. **About the evaluation protocol (Reviewer *e3A6*)**
>
>     We found that we misunderstood the hyperparameter search process, and the experimental results were misreported due to the misunderstanding, while we prepared the author responses. Not only the model selection but also the hyperparameter search should be conducted on the training-domain validation set in DomainBed. However, we reported our GESTUR’s performance scores with the hyperparameters searched on the testing-domain validation set. We update the performance scores and analysis with the hyperparameters searched on the training-domain validation set for a fair comparison, and modify the manuscript of the evaluation protocol. We are deeply sorry again for the inconvenience.
>
> We will respond to each reviewer's comments below.
>
> ---
> **References**
>
> [1] Ensemble of Averages: Improving Model Selection and Boosting Performance in Domain Generalization, Arpit et al., NeurIPS (2022)

---

### Author Response · Authors · 2022-11-24
**The end of the rebuttal phase approaching**

Dear Reviewers,

We gently remind you that the rebuttal period is about to close. Could you please go over our responses and the revision of our paper? We have responded to your comments and faithfully reflected them in the revision. And, we also provided the additional experimental results that you requested. We sincerely thank you for your insightful and constructive comments on our paper.

Thank you

---

### Decision · Program_Chairs · 2023-01-20

**Decision:**

Reject

**Justification For Why Not Higher Score:**

A major concern the reviewers have about this paper is the feasibility and reliability of using large pretrained models as a loose approximation of arbitrary unseen domains for the purpose of gradient estimation. They also have reservations on the novelty of the work.

**Justification For Why Not Lower Score:**

NA

**Metareview: Summary, Strengths And Weaknesses:**

Fine-tuning a large pretrained model on a downstream task is common practice.  The fine-tuning takes place on data from a source domain and the fine-tuned model will be applied to a target domain.  To improve the model performance on the target domain, this paper proposes to use, during fine-tuning,  the gradients of the unseen target domain, which are estimated using the pretrained model. A major concern the reviewers have about this paper is the feasibility and reliability of using large pretrained models as a loose approximation of arbitrary unseen domains for the purpose of gradient estimation. They also have reservations on the novelty of the work.